# Vasoactive pharmacological management according to SCAI class in patients with acute myocardial infarction and cardiogenic shock

Nanna Louise Junker Udesen[1]*, Ole Kristian Lerche Helgestad[1], Jakob Josiassen[2], Christian Hassager[2], Henrik Frederiksen Højgaard[3], Louise Linde[1], Jesper Kjaergaard[2], Lene Holmvang[2], Lisette Okkels Jensen[1], Henrik Schmidt[3], Hanne Berg Ravn[3], Jacob Eifer Møller[1,2]

1 Department of Cardiology, Odense University Hospital, Odense, Denmark, 2 Department of Cardiology, Copenhagen University Hospital, Rigshospitalet, Copenhagen, Denmark, 3 Department of Cardiothoracic Anesthesia, Odense University Hospital, Odense, Denmark

* Nanna.louise.junker.udesen@rsyd.dk

**Data Availability Statement:** It is not possible to give public access to the data as this would be against the Danish legislation. Permission to data

## Abstract

### Background

Vasoactive treatment is a cornerstone in treating hypoperfusion in cardiogenic shock following acute myocardial infarction (AMICS). The purpose was to compare the achievement of treatment targets and outcome in relation to vasoactive strategy in AMICS patients stratified according to the Society of Cardiovascular Angiography and Interventions (SCAI) shock classification.

### Methods

Retrospective analysis of patients with AMICS admitted to cardiac intensive care unit at two tertiary cardiac centers during 2010–2017 with retrieval of real-time hemodynamic data and dosages of vasoactive drugs from intensive care unit databases.

### Results

Out of 1,249 AMICS patients classified into SCAI class C, D, and E, mortality increased for each shock stage from 34% to 60%, and 82% (p<0.001). Treatment targets of mean arterial blood pressure > 65mmHg and venous oxygen saturation > 55% were reached in the majority of patients; however, more patients in SCAI class D and E had values below treatment targets within 24 hours (p<0.001) despite higher vasoactive load and increased use of epinephrine for each severity stage (p<0.001). In univariate analysis no significant difference in mortality within SCAI class D and E regarding vasoactive strategy was observed, however in SCAI class C, epinephrine was associated with higher mortality and a significantly higher vasoactive load to reach treatment targets. In multivariate analysis there was no statistically association between individually vasoactive choice within each SCAI class and 30-day mortality.

can be granted from the Danish Data Protection Agency (contact via nanna.louise.junker. udesen@rsyd.dk or dt@datatilsynet.dk) for researchers who meet the criteria for access to confidential data.

**Funding:** JEM received a research grant from Abiomed and the Danish Heart association. NLJU received funding from the University of Southern Denmark and Region of Southern Denmark. The funders had no role in study design, data collection and analysis, decision to publish, or preparation of the manuscript

**Competing interests:** Dr. JE. Møller has received a research grant from Abiomed. Dr. OKL Helgestad and Dr. NLJ Udesen have received travel compensation from Abiomed. These competing interests does not alter our adherence to PLOS ONE policies on sharing data and materials. The remaining authors have no disclosures

## Conclusion

Hemodynamic treatment targets were achieved in most patients at the expense of increased vasoactive load and more frequent use of epinephrine for each shock severity stage. Mortality was high regardless of vasoactive strategy; only in SCAI class C, epinephrine was associated with a significantly higher mortality, but the signal was not significant in adjusted analysis.

## Introduction

Approximately 5–10% of patients suffering an acute myocardial infarction will develop cardiogenic shock (AMICS) characterized by inadequate cardiac output to meet the oxygen demand of the body leading to organ hypoperfusion, a condition associated with a short-term mortality of approximately 50% [1, 2]. Immediate revascularization is the only treatment proven to reduce mortality in AMICS; besides this, limited evidence directs the way through improved outcome [3, 4].

Treatment with inopressors, typically catecholamines, is used in 70–94% of AMICS cases to reverse hypotension and organ hypoperfusion by increasing systemic vascular resistance and improving cardiac output (CO) by increasing cardiac contractility [1, 5, 6]. Guidelines recommend using norepinephrine (NE) as first inopressor for hypotension in AMICS [7, 8], primarily based on SOAP-II trial comparing first line NE and dopamine and the CAT study comparing first line NE with epinephrine [9, 10]. Both studies studied mixed populations of critically ill patients and even though the studies were neutral on primary endpoint (30 day mortality), safety concerns were raised which is in agreement with smaller randomized controlled trials and observational studies suggesting more harm with epinephrine and dopamine [11, 12]. Inodilators may be given to augment CO and decrease afterload if this is tolerated [13]. Few randomized controlled trials exist in the field of vasoactive support in AMICS, and since the level of support in AMICS often is guided by hemodynamic and metabolic state [4, 8], knowledge from observational studies is prone to confounding by indication. Observational studies clearly demonstrate that higher doses of inopressors or use of epinephrine are associated with higher mortality [6, 14, 15]; however, such studies often assume homogeneity in the AMICS population and do not stratify according to the severity of the disease.

The Society of Cardiovascular Angiographic and Interventions (SCAI) has developed a shock classification to define some distinct phenotypes of cardiogenic shock (CS) [16, 17]. Analysis of vasoactive treatment within SCAI subgroups of AMICS can potentially lead to a more nuanced insight into vasoactive choice and treatment effect.

The current study aimed to investigate the characteristics, achievement of hemodynamic goals, and outcome related to vasoactive choice in SCAI shock class C, D, and E among patients with AMICS.

## Methods

### Population

The present study is based on the Retroshock cohort, consisting of 1,716 patients admitted with AMICS at two tertiary cardiac centers (Rigshospitalet, Copenhagen and Odense University Hospital, Odense, Denmark) from 2010 to 2017. These centers provide tertiary cardiac care for a population of 3.8 million people. A detailed description of the overall study

population has been published previously [1]. The study was approved by the Danish Patient Safety Authority (ID: 3-3013-1133/1) and the Danish Data Protection Agency (ID: 16/7381 and 18/23756).

AMICS was defined as presence of myocardial infarction (based on the fourth Universal Definition of Myocardial Infarction [18]), and CS was defined as presence of (all required).

a. hypotension with systolic blood pressure < 90 mmHg or treatment with either inopressors or mechanical circulatory support (MCS)

b. signs of end-organ hypoperfusion (cold/clammy skin or oliguria or altered mental status or arterial lactate >2.5 mmol/L)

c. reduction in left or right ventricular function due to AMI.

In addition, only patients admitted to cardiac intensive care unit (CICU) where vasoactive treatment (inopressor and/or inodilator) was given as continuous infusion were eligible for the present study (N = 1,249), Fig 1. Further, patients were excluded if they 1) did not survive to CICU admission, 2) were treated with infusion of inopressors at a referring hospital before admission to treating center, or 3) if they were transferred to a general ICU.

## SCAI classification

The SCAI shock classification was interpreted from the consensus statement by the Society of Cardiovascular Angiography and Interventions [16]. Patients were divided in SCAI class C to E according to Table 1, no patients in the study were considered in SCAI class A or B.

## Vasoactive drugs

The choice of individual vasoactive drugs was at the discretion of the attending physicians. Generally, hemodynamic targets were mean arterial pressure (MAP) of 65 mmHg or more and central venous oxygenation (ScVO$_2$) of 55% or more. Inopressors for the first 72 hours were classified according to drug type, dose, and whether it was given alone or in combination. Inopressor regimes were divided into the following groups: norepinephrine (NE), dopamine (DA), combined use of DA and NE (NE+DA), and addition of epinephrine to DA and/or NE

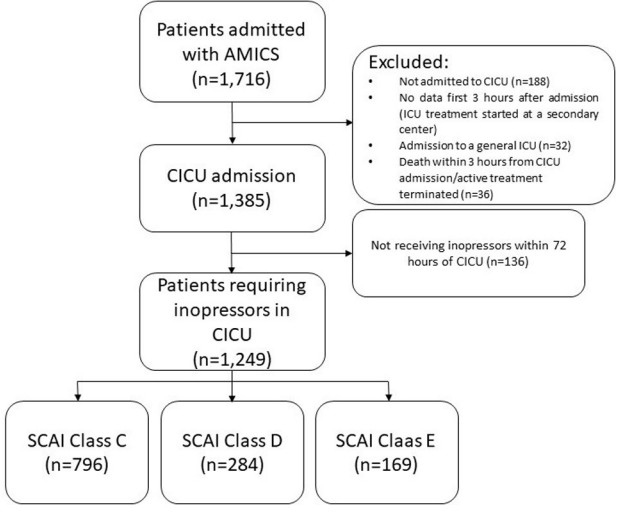

**Fig 1. Consort diagram.**

Table 1. Modified SCAI class definition.

| SCAI C (Classic) | SCAI D (Deteriorating) | SCAI E (Extreme) |
|---|---|---|
| Verified AMICS reaching CICU and requering vasopressors < 72 hours | | |
| Hypoperfusion with requirement of vasopressor treatment | First CICU arterial lactate > 2 mmol/L and persistent hypoperfusion, with | Severe hypoperfusion with |
| | a) Arterial lactate increase of $\geq$ 0.5 mmol/L < 24 hours after CICU admission, and/or<br>b) MAP < 65 mmHg the first 6 hours after CICU admission | a) Use of VA-ECMO < 24 hours after CICU admission<br>b) Arterial lactate $\geq$ 10 mmol/L <24 hours after CICU admission |

VA-ECMO: veno-arterial extracorporeal membrane oxygenation, MAP: mean arterial blood pressure.

(MIX+AD). Administration of dobutamine, milrinone, or levosimendan within 24 hours after arrival at CICU was classified as early inodilator treatment. No patients were treated with phenylephrine infusion.

To quantify the total amount of combined pharmacologic cardiovascular support the Vasopressor-inotropic-score (VIS) was calculated from the formula: dopamine (ug/kg/min) + dobutamine (ug/kg/min) + 100 x epinephrine (ug/kg/min) + 100 x norepinephrine (ug/kg/min) + milrinone x 10 (ug/kg/min) + 50 x levosimendan (ug/kg/min) + 10.000 x vasopressin (U/kg/min) [19].

## Data collection

Demographic data and admission characteristics were obtained from a detailed chart review of each patient after all patients had been identified through the Danish National Patient Registry, and the AMICS diagnosis had been validated through chart review [1]. Coronary intervention data was collected from the Western Denmark Heart Registry (WDHR) and Eastern Denmark Heart Registry (PATS). Real-time hemodynamic parameters were obtained from the CICU databases (Picis clinical solutions and Intellispace Critical Care & Anesthesia). CICU data included continuous recordings of the infusion rate of administrated drugs, hemodynamic measurements, and blood gases obtained from arterial line, central venous catheter, and pulmonary catheter if placed.

Baseline was selected as a timeframe of 3 hours from CICU arrival to allow for installation at the CICU, placement of invasive catheters, and any delay in recording of variables. Subsequent data were collected at 6, 9, 12, 15, 18, 24, and 48 hours after CICU admission. If a parameter was recorded more than once in one hour, the mean was calculated.

## Statistics

Continuous variables are presented as mean with standard deviation (SD) or median with interquartile range of 25th and 75th (IQR) depending on the variables followed a Gaussian distribution. Categorical variables are presented as frequencies (percentages). Comparison between groups was analysed using the Student's t-test, or the nonparametric Kruskal-Wallis test for continuous variables and the $chi_2$-test for categorical variables. Odds ratios were calculated with logistic regression analysis. First unadjusted analyses were performed and subsequently after adjusting for age, OHCA, lactate at CICU admission, revascularization, use of advanced MCS, maximum VIS and renal replacement therapy. Repeated measures were analysed with a mixed linear model with SCAI class C and baseline as reference. Graphs over time illustrate mean values with 95% confidence interval (95%CI).

All analyses were performed by using STATA statistical software (Stata Corp LLC). The significance level was set at $p < 0.05$.

## Results

Of 1,249 included AMICS patients, 796 (64%) of patients were classified as SCAI class C, 284 patients (23%) as SCAI class D, and 169 (13%) as SCAI class E (Fig 1). Comorbidities and admission characteristics are shown in Table 2 for each SCAI class. Overall, there were no significant differences in comorbidities other than diabetes being more frequent in SCAI class E, and among SCAI class D patients, mean age was higher with a history of stroke being more prevalent (Table 2).

Out-of-hospital cardiac arrest occurred in 57% of patients in SCAI C vs. 41% in SCAI D and E (p<0.001), and index systolic blood pressure was higher and heart rate lower among SCAI C patients (p<0.001). Overall metabolic profile at arrival was more compromised in SCAI class E patients with lower blood pH, lower bicarbonate, and higher blood glucose (p<0.001), Table 2. As expected, arterial lactate at arrival increased according to SCAI class (Table 2). Immediate revascularization was performed in the majority of patients irrespective

**Table 2. Baseline characteristics of 1,249 AMICS patients with inopressor requirements at CICU.**

|  | SCAI C | SCAI D | SCAI E | *p-value* |
|---|---|---|---|---|
|  | N = 796 | N = 284 | N = 169 |  |
| Age, years | 64 (11) | 70 (10) | 64 (12) | <0.001 |
| Male gender | 648 (81%) | 204 (72%) | 128 (76%) | 0.002 |
| History of |  |  |  |  |
| Hypertension | 379 (49%) | 151 (56%) | 73 (47%) | 0.11 |
| Ischemic heart disease | 205 (26%) | 90 (32%) | 49 (31%) | 0.11 |
| Myocardial infarction | 110 (14%) | 48 (17%) | 22 (14%) | 0.39 |
| Diabetes |  |  |  | 0.03 |
| No diabetes | 643 (84%) | 217 (81%) | 115 (74%) |  |
| Diabetes type 1 | 15 (2%) | 6 (2%) | 2 (1%) |  |
| Diabetes type 2 | 112 (14%) | 45 (17%) | 39 (25%) |  |
| Peripheral arterial disease | 49 (6%) | 29 (11%) | 13 (8%) | 0.061 |
| COPD | 71 (9%) | 33 (12%) | 15 (10%) | 0.36 |
| Stroke | 56 (7%) | 31 (11%) | 3 (2%) | 0.002 |
| Characteristics at arrival |  |  |  |  |
| OHCA | 451 (57%) | 116 (41%) | 70 (41%) | <0.001 |
| LVEF, % | 30 (20, 40) | 30 (20–35) | 20 (10–40) | <0.001 |
| Heart rate, min$^{-1}$ | 82 (69–99) | 86 (74–102) | 90 (75–105) | 0.004 |
| Systolic BP, mmHg | 85 (77–94) | 84 (74–94) | 80 (70–90) | <0.001 |
| Arterial lactate, mmol/l | 4.4 (2.5–7.8) | 5.3 (3.7–8.4) | 11.9 (8.0–14.3) | <0.001 |
| PH | 7.30 (7.24–7.35) | 7.25 (7.18–7.31) | 7.17 (7.08–7.27) | <0.001 |
| HCO$_3$, mmol/l | 20.3 (18.5–22) | 18.7 (16.5–20.4) | 16.6 (13.9–19.7) | <0.001 |
| Glucose, mmol/l | 9.7 (7.7–12.8) | 12.5 (10.0–15.2) | 14.2 (9.6–19.3) | <0.001 |
| Revascularization | 726 (91%) | 246 (87%) | 153 (91%) | 0.083 |
| Coronary culprit lesion |  |  |  | <0.001 |
| Left Main | 54 (7%) | 38 (15%) | 35 (23%) |  |
| LAD | 337 (46%) | 110 (45%) | 62 (41%) |  |
| LCx | 131 (18%) | 27 (11%) | 10 (7%) |  |
| RCA | 204 (28%) | 71 (29%) | 46 (30%) |  |

Data are presented as mean with standard deviation (SD) or median with interquartile range of 25$^{th}$ and 75$^{th}$ (IQR). COPD: Chronic obstructive pulmonary disease; OHCA: out-of-hospital cardiac arrest; LVEF: left ventricular ejection fraction; BP: blood pressure, LAD: left anterior descending artery, LCx: left circumflex artery, RCA: right coronary artery.

of SCAI class and vasoactive strategy (Tables 2 and S2). However, the culprit lesion was more frequently located in the left main coronary artery in SCAI E patients (Table 2). In relation to the choice of vasoactive strategy within each SCAI group, there were pronounced differences in patient characteristics in SCAI class C, while there were minor differences in the more severe SCAI stages (S2 Table).

## Vasoactive support at CICU

Infusion of inopressors to achieve blood pressure target was initiated in all patients with significant differences in choice and dosages of inopressor according to SCAI classes (Table 3 and Fig 2). Hemodynamic goals, including MAP and especially ScVO$_2$, were achieved in the majority of patients (S1 Fig), although an increasingly proportion in SCAI class D and E, had MAP below 65 mmHg and ScVO$_2$ below 55% within the first 24 hours (Table 3). VIS to attain treatment targets remained significantly higher in SCAI class D and E throughout the first days of CICU course, demonstrated by a lower MAP/ VIS ratio (Fig 3). Generally, VIS over time differed significantly depending on inopressor choice in each SCAI class (Figs 4 and S2).

Irrespective of SCAI class, NE infusion was the preferred inopressor, but dosage was twice as high in SCAI class D and almost three times higher in SCAI class E compared to SCAI class C (Table 3 and Fig 4). Epinephrine infusion was most frequent in SCAI class E patients, 65% vs. 11% and 40% in SCAI class C and D patients, and if used, epinephrine dosages were

**Table 3. Findings at CICU and support during CICU course.**

|  | SCAI C | SCAI D | SCAI E | p-value |
|---|---|---|---|---|
| **Initial presentation at CICU:** |  |  |  |  |
| Heart rate, min$^{-1}$ | 81 (67–96) | 90 (78–103) | 96 (82–108) | <0.001 |
| Arterial lactate, mmol/L | 2.1 (1.4–3.7) | 4.0 (2.8–5.5) | 10.4 (5.9–12.7) | <0.001 |
| First VIS | 6 (3–13) | 15 (5–29) | 23 (10–42) | <0.001 |
| Systolic BP, mmHg | 98 (91–108) | 90 (79–102) | 82 (73–95) | <0.001 |
| Mean BP, mmHg | 72 (67–78) | 67 (60–74) | 64 (57–73) | <0.001 |
| Percentages of time < 24 H with MAP < 65 mmHg | 19% | 34% | 40% | <0.001 |
| ScVO$_2$, oxygen % | 69 (60–75) | 65 (56–74) | 63 (54–70) | <0.001 |
| Percentages of time < 24 H with ScVO$_2$ < 55% | 4% | 7% | 8% | <0.001 |
| Mechanical ventilation | 712 (90%) | 259 (91%) | 164 (97%) | 0.011 |
| Impella | 95 (12%) | 40 (14%) | 50 (30%) | <0.001 |
| VA-ECMO | 4 (1%) | 1 (0%) | 45 (27%) | <0.001 |
| IABP | 87 (11%) | 29 (10%) | 26 (15%) | 0.22 |
| Inodilators < 24 H | 191 (24%) | 111 (39%) | 77 (46%) | <0.001 |
| Norepinephrine | 645 (81%) | 247 (87%) | 160 (95%) | <0.001 |
| Dopamine | 563 (71%) | 164 (58%) | 83 (49%) | <0.001 |
| Epinephrine | 90 (11%) | 115 (40%) | 110 (65%) | <0.001 |
| Maximum doses |  |  |  |  |
| Norepinephrine, ug/kg/min | 0.13 (0.07–0.23) | 0.25 (0.15–0.38 | 0.35 (0.22–0.5) | <0.001 |
| Dopamine, ug/kg/min | 7 (4–10) | 6 (4–10) | 7 (4–10) | 0.52 |
| Epinephrine, ug/kg/min | 0.08 (0.04–0.18) | 0.15 (0.07–0.25) | 0.17(0.1–0.34) | <0.001 |
| VIS maximum < 24 hours | 15 (9–26) | 30 (17–50) | 50 (31–72) | <0.001 |

Data are presented as frequencies with percentages (%) or median with interquartile range of 25$^{th}$ and 75$^{th}$ (IQR). VIS: vasopressor-inotropic-score; ScVO$_2$: central venous oxygen saturation; BP: blood pressure; CRRT: continuous renal replacement therapy; VA-ECMO: veno-arterial extracorporeal membrane oxygenation; IABP: intra aorta balloon pump; max dose: maximum dosage; H: hours.

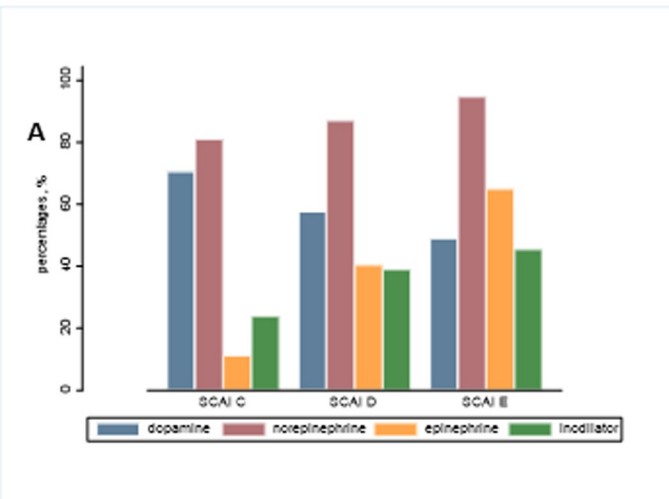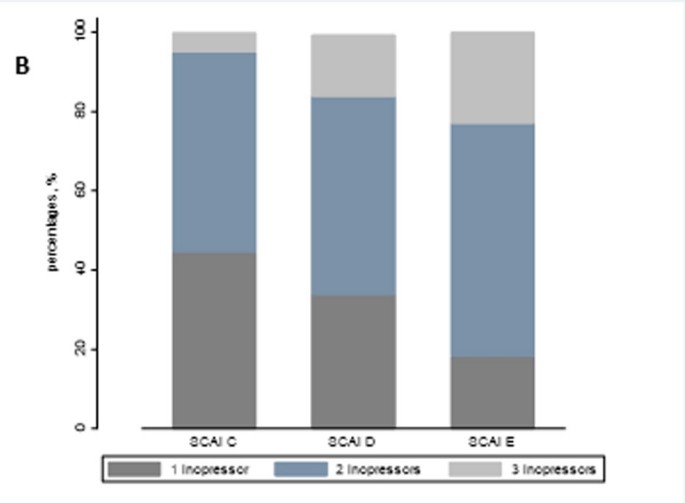

**Fig 2.** A) Vasoactive distribution in each SCAI class B) Number of inopressors according to SCAI class.

considerably higher among SCAI class D and E patients (Table 3). Dopamine was less frequently used in SCAI class E, but doses of dopamine did not differ between groups (Table 3). Combinations of inopressors were more frequent among SCAI class D and E patients, where patients often required more than one inopressor (Table 4 and Fig 2). Higher inopressor support for patients in SCAI class D and E were reflected by higher VIS already at CICU admission, despite lower systolic blood pressure (Table 3).

The occurrence of ventricular arrhythmias was less frequent in SCAI class C patients, and in general, there was no significant association between ventricular arrhythmias and inopressor choice in each SCAI class (Table 4). There was no significant difference in atrial arrhythmias between SCAI classes (Table 3), but in both SCAI class C and D, the incidence of atrial arrhythmias was low among only dopamine treated, while epinephrine was associated with slightly increased frequency of atrial arrhythmias in SCAI class C (Table 3). Use of epinephrine was associated with increased requirement of continuous renal replacement therapy (CRRT)

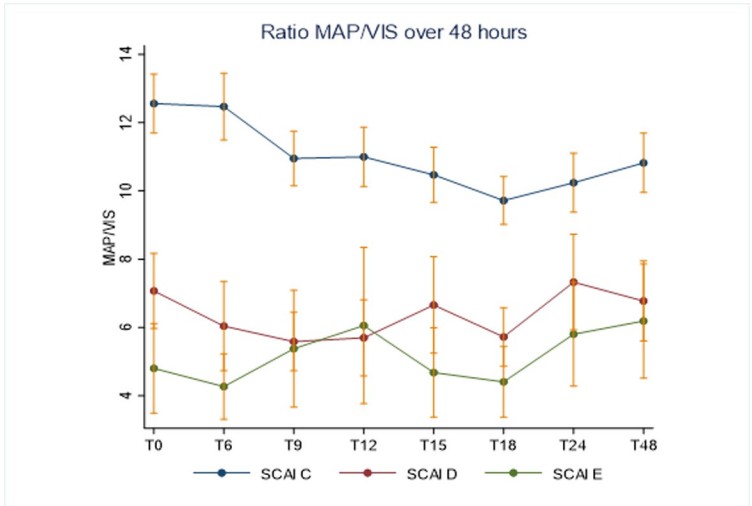

**Fig 3. Ratio MAP / Vasopressor-inotropic score over 48 hours.**

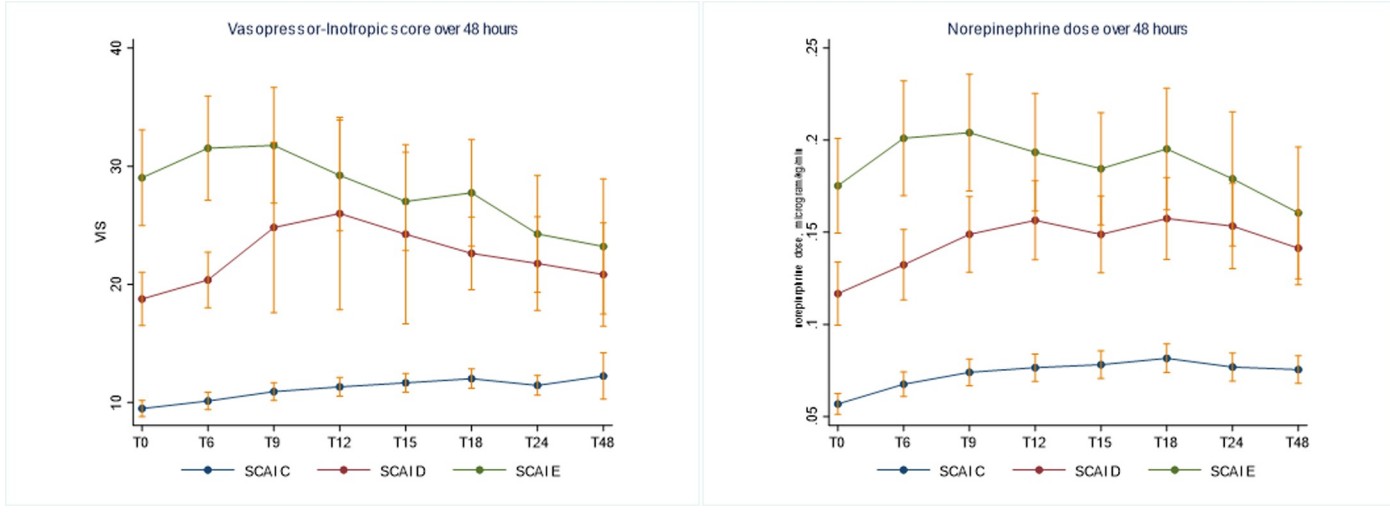

**Fig 4.** A) Vasopressor-Inotropic score and B) norepinephrine dose overtime each in SCAI class.

**Table 4. Vasoactive treatment and association with arrhythmias, renal replacement therapy and 30-day mortality in each SCAI class.**

| | TOTAL | NE | DA | NE+DA | MIX+AD | P | NO INODILATOR | INODILATOR | P |
|---|---|---|---|---|---|---|---|---|---|
| **SCAI CLASS C** | N = 796 | N = 197 (25%) | N = 142 (18%) | N = 367 (46%) | N = 90 (11%) | | N = 605 (76%) | N = 191 (24%) | |
| VIS MAXIMUM | 15 (9–26) | 14 (8–25) | 6 (5–9) | 18 (12–26) | 37 (23–54) | <0.001 | 13 (8–22) | 24 (14–37) | <0.001 |
| CRRT | 19% (152) | 17% | 4% | 17% | 53% | <0.001 | 15% | 33% | <0.001 |
| ATRIAL ARRHYTHMIA | 25% (201) | 25% | 17% | 27% | 32% | 0.04 | 21% | 39% | <0.001 |
| VENTRICULAR ARRHYTHMIA | 26% (207) | 23% | 20% | 28% | 34% | NS | 23% | 36% | 0.001 |
| DEATH < 30 DAYS | 34% (267) | 28% | 25% | 36% | 51% | <0.001 | 33% | 36% | NS |
| **SCAI CLASS D** | N = 284 | N = 63 (22%) | N = 23 (8%) | N = 83 (29%) | N = 110 (39%) | | N = 173 (61%) | N = 111 (39%) | |
| VIS MAXIMUM | 30 (17–50) | 25 (15–31) | 5 (3–9) | 27 (16–38) | 48 (30–64) | <0.001 | 25 (12–40) | 38 (26–61) | <0.001 |
| CRRT | 31% (88) | 21% | 4% | 31% | 44% | <0.001 | 25% | 41% | 0.005 |
| ATRIAL ARRHYTHMIA | 28% (78) | 31% | 4% | 30% | 31% | NS | 23% | 35% | 0.023 |
| VENTRICULAR ARRHYTHMIA | 34% (95) | 35% | 17% | 33% | 36% | NS | 30% | 40% | NS |
| DEATH < 30 DAYS | 60% (170) | 63% | 43% | 59% | 60% | NS | 64% | 53% | NS |
| **SCAI CLASS E** | N = 169 | N = 25 (15%) | N = 2 (1%) | N = 32 (19%) | N = 107 (63%) | | N = 92 (54%) | N = 77 (46%) | |
| VIS MAXIMUM | 50 (32–72) | 25 (17–44) | | 34 (16–56) | 57 (41–80) | <0.001 | 44 (26–69) | 54 (36–74) | NS |
| CRRT | 55% (93) | 56% | 0 | 50% | 59% | NS | 49% | 62% | NS |
| ATRIAL ARRHYTHMIA | 30% (50) | 36% | 50% | 23% | 30% | NS | 22% | 41% | 0.008 |
| VENTRICULAR ARRHYTHMIA | 35% (59) | 32% | 0 | 50% | 32% | NS | 29% | 42% | NS |
| DEATH < 30 DAYS | 82% (139) | 72% | 50% | 84% | 84% | NS | 83% | 82% | NS |

Data presented as median (IQR) and percentages (%). NE: norepinephrine, DA: dopamine, MIX+AD: addition of epinephrine to dopamine and/ or norepinephrine. CRRT: continues renal replacement therapy. Eight Tpatients (5 in SCAI class D and 3 in SCAI class E) who received epinephrine alone are counted in the total number but excluded on the group level, due to low number.

in both SCAI class C and D, but among SCAI class E patients, CRRT treatment was equally high independent of inopressor choice (Table 4).

Inodilators were initiated in 379 patients (30%) within 24 hours of admittance and more frequently in SCAI class E patients, 46% vs. 24% and 39% in SCAI class C and SCAI class D, respectively. Across all SCAI groups, inodilator treated patients more frequently required CRRT and developed atrial arrhythmias more often (Table 4). Ventricular arrhythmias tended to be more frequent among inodilator treated patients, which were significant for patients in SCAI class C (Table 4).

## 30 days mortality

Death within 30 days occurred in 34%, 60%, and 82% of patients according to SCAI class C, D, and E with different causes of death (Tables 4 and S3), and in SCAI class E, more than half of patients had died at day three (S1 Table). There was no significant difference in mortality concerning inopressor regime in SCAI class D and E despite higher VIS for patients receiving combinations of inopressors, but in SCAI class C, mortality was significantly higher in patients given epinephrine infusion (Table 4). The higher mortality among epinephrine treated in SCAI class C was not statically significant after adjusting for initial arterial lactate, revascularization, advanced MCS and renal replacement therapy (Table 5 and S5 Fig).

## Discussion

This is the first explorative study to describe vasoactive strategies at different severity degrees of AMICS based on the SCAI classification. Among 1,249 patients categorized as classic (C), deteriorating (D), or extreme (E) manifestation of AMICS, the doses of inopressors, except for dopamine, increased with shock severity, and epinephrine became almost six times more prevalent in the extreme cases, where also inodilator treatment was most frequent. The severity of AMICS increased for each shock stage with increasing mortality, even though the majority of

**Table 5. Logistic regression on 30-days mortality.**

| death30 | 30-day mortality Unadjusted | | | | 30-day mortality Adjusted* | | | |
|---|---|---|---|---|---|---|---|---|
| | Odds ratio | [95% CI] | | P-value | Odds ratio | [95% CI] | | P-value |
| **Epinephrine** | | | | | | | | |
| **SCAI C** | 2.29 | 1.47 | 3.57 | 0.000 | 1.28 | 0.72 | 2.28 | 0.396 |
| **SCAI D** | 1.06 | 0.65 | 1.73 | 0.814 | 0.70 | 0.37 | 1.31 | 0.364 |
| **SCAI E** | 1.50 | 0.67 | 3.35 | 0.326 | 0.78 | 0.27 | 2.27 | 0.646 |
| **Norepinephrine** | | | | | | | | |
| **SCAI C** | 1.51 | 1.01 | 2.24 | 0.042 | 0.78 | 0.49 | 1.25 | 0.30 |
| **SCAI D** | 1.32 | 0.63 | 2.76 | 0.463 | 0.64 | 0.23 | 1.83 | 0.41 |
| **SCAI E** | 0.90 | 0.10 | 8.03 | 0.927 | 1.16 | 0.11 | 12.90 | 0.9 |
| **Dopamine** | | | | | | | | |
| **SCAI C** | 1.12 | 0.81 | 1.55 | 0.493 | 1.16 | 0.80 | 1.66 | 0.44 |
| **SCAI D** | 0.83 | 0.51 | 1.35 | 0.460 | 0.99 | 0.55 | 1.78 | 0.981 |
| **SCAI E** | 0.85 | 0.38 | 1.88 | 0.687 | 0.64 | 0.25 | 1.66 | 0.362 |
| **Inodilator** | | | | | | | | |
| **SCAI C** | 1.16 | 0.83 | 1.63 | 0.386 | 0.93 | 0.62 | 1.40 | 0.723 |
| **SCAI D** | 0.65 | 0.39 | 1.05 | 0.079 | 0.62 | 0.34 | 1.12 | 0.111 |
| **SCAI E** | 0.99 | 0.45 | 2.18 | 0.973 | 1.39 | 0.53 | 3.68 | 0.506 |

*adjusted for age, arterial lactate at CICU arrival, OHCA, revascularization, advanced MCS, maximum VIS and renal replacement therapy.

patients achieved treatment goals with MAP > 65 mmHg and venous oxygen tension > 55%. However, even with a high vasoactive load in SCAI class D and E, the response in blood pressure and venous oxygen saturations were less pronounced.

AMICS is a heterogeneous condition, where some patients survive with a low level of circulatory support while other AMICS patients in refractory shock will be resistant even to aggressive mechanical circulatory support and will develop irreversible multiorgan failure [4]. Hence, the SCAI classification has been proposed to classify AMICS in subgroups depending on the severity [16]. In agreement dividing the current cohort in SCAI class C to E identified patients with a very different outcome.

The current study found no difference in mortality related to different inopressor strategies within SCAI classes D and E, only epinephrine use among SCAI class C was associated with increased mortality in the unadjusted analysis but not in adjusted analysis. Epinephrine use is commonly associated with detrimental outcomes in observational studies, even after propensity matching [6, 14, 20]. However, like the present study, there is an inherent selection bias in observational studies as epinephrine seldom is the first-line treatment in AMICS but often used in patients not responding to initial inopressor treatment thus introducing a confounding by indication [21]. Hence, in the present study, the vasoactive load among epinephrine-treated patients was higher than any other inopressor strategy. In a large CICU population where VIS were comparable for NE and epinephrine, Jentzer et al. found no association between epinephrine and mortality [22] which also is in agreement with the randomized CAT study comparing first line NE in a mixed population of hypotensive critically ill patients. Opposed to this, Levy et al. demonstrated in a randomized study comparable effects on cardiac index but a higher incidence of refractory AMICS with epinephrine compared to NE in 57 AMICS patients [11]. Lactate acidosis was presumable the primary contributor in the definition of refractory AMICS as no difference in other hemodynamic variables or in kidney and liver parameters were observed between the treatments. Thus the premature termination of the study may have been influenced by known effects of epinephrine, as $\beta_2$-adrenergic accelerated glycolysis induced by epinephrine likely have contributed to lactate accumulation in some patients [11, 23]. Notably increase lactate was frequent in the CAT study during initial 24 hours and reason for discontinuation of study drug in 12% of epinephrine treated patients.

The overall occurrence of arrhythmias requiring intervention was lower among dopamine treated, which is in contrast to the main findings in the landmark SOAP-II trial, where dopamine was found to be arrhythmogenic and cause tachycardia compared to NE in equipotent doses across all types of shock [9]. The median dose of dopamine in SOAP-II was 16 μg/kg/min during the initial phase opposed to the present study, where doses exceeding 10 μg/kg/min were infrequent. Mortality was generally lower among dopamine-treated individuals, in accordance with these patients being in less severe shock not requiring a high vasoactive load or MCS to achieve hemodynamic goals. Hence, dopamine alone was rarely used in SCAI D and E. This emphasizes that low to moderate doses of dopamine is feasible and safe in selected patients who might require chronotropic support (S3 Fig), but should be shifted or combined with NE at higher dosages due to the risk of arrhythmias during administration of high doses [9, 24].

Inodilators aim to provide afterload reduction and concomitantly increase cardiac contractility, thus counterbalancing the excessive vasoconstriction caused by inopressors, thereby improving CO [13]. In a recent study of CS of various etiologies, dobutamine and milrinone were equally effective [13]. There are no similar data for levosimendan in CS, but in acute decompensated heart failure, levosimendan failed to be superior to dobutamine [25]. In the current study, inodilators were more commonly used in SCAI class D and E, presumably reflecting a more critical need of increasing CO in these stages, and generally in patients with

high occurrence of acute kidney injury and associated with more arrhythmias. In a recent meta-analysis, the use of early inodilators were associated with lower mortality among CS patients receiving inopressors [26]. However, a small randomized controlled trial of 30 CS patients found no difference in global hemodynamic changes whether patients received epinephrine or NE and dobutamine [27]. The present study found no significant survival benefit with inodilators, but mortality was non-statistical lower in SCAI D and E despite signs of more advanced disease.

In the current study, choice and dosage of vasoactive drugs were at the discretion of the attending physician and generally targeted towards achieving MAP>65 mmHg to avoid cerebral and coronary ischemia, and $ScVO_2$ >55% to ensure oxygen delivery. In general, these targets were achieved, however among the extreme cases; more patients did not reach MAP target despite much higher dosages and number of drugs. Except for epinephrine-treated patients with a high vasoactive load to maintain MAP, in SCAI class C, there was no significant difference in mortality within the individual SCAI classes. However, the heterogeneity in disease presentation based on vasoactive strategy was extensive, especially in SCAI class C, where the subgroup that received epinephrine had signs of more advanced disease based on the patient characteristics. Although speculative, this could suggest that the outcome to a higher degree is determined by disease severity than the individual vasoactive strategy, supported by the significant increase in mortality disappeared after adjusting for other factors known to be associated with severity degree. To extrapolate these findings into future use, adequately sized randomized controlled trials on vasoactive agents in AMICS ideally stratified by SCAI class are essential. In retrospective studies, it is necessary to nuance whether a treatment is chosen as the last attempt when others have failed, and therefore will be associated with mortality or if the treatment in itself leads to poor outcome. Further, the accumulated vasoactive load should be described in comparisons of vasoactive strategies.

The optimal inopressor in the treatment of AMICS should provide the best hemodynamic support at the lowest cost in terms of myocardial energy consumption as well as avoiding excessive vasoconstriction leading to ischemia and arrhythmia. Tachycardia potentiates arrhythmias, and as both epinephrine and dopamine act chronotropic, NE for the homogeneous group of AMICS is probably safer as it has less effect on heart rate [28]. However at a high level of vasoconstriction and hypoperfusion, inotropes cannot be avoided, which often will increase heart rate and invariably increase myocardial oxygen consumption. Based on severity degree and hemodynamics, low dose dopamine may be considered for bradycardia and epinephrine for the severe stages. It is difficult to conclude which strategy should be preferred at deteriorating state, but in current study where epinephrine properly was given as rescue therapy there was no difference in mortality in deteriorating or extreme cases in regards to vasoactive strategy.

## Limitations

The most important limitation is the retrospective design, which is prone to selection bias and confounding by indication. Also, the numbers were limited in each vasoactive group after SCAI division, increasing the risk of type 2 error. The associations found in the current study cannot prove any causality but only describe what we observed by dividing the AMICS population according to severity degree. In SCAI class D and E, mean VIS declined after 12 hours (Fig 4); this can be explained by early mortality among patients with the highest VIS in SCAI D, as the decline became less pronounced after extracting patients who died within 48 hours (S4 Fig). In SCAI class E, the decline in VIS stayed after extraction of those who died within 48 hours, why it could reflect lowering of VIS after the establishment of MCS in some. The SCAI

criteria were chosen from changes in blood pressure, lactate levels, and early VA-ECMO within 24 hours, which were an interpretation of the proposed SCAI division; however, the interpretation of deteriorating and extreme manifestations are not unambiguous and differ among studies [29, 30]. We did not include VIS in the SCAI classification, as this would cause confounding in relation to analysing inopressors in the respective SCAI classes. However, no classification tools in critically ill populations are flawless, and it can be argued that there may have been patients in SCAI class C among the epinephrine treated who more likely belonged to SCAI class D but were missed by the SCAI criteria. Across cohorts, there are quite divergent proportions in the SCAI groups, and outcome varies [29–31], which probably reflect differences in cohorts, different interpretations, and definitions of the SCAI criteria. Further, if vasoactive support is a prerequisite in SCAI C, this is sensitive to the classification, as vasoactive use varies across countries [32]. In the current study, patients were excluded if they were not admitted to CICU or died within 3 hours after CICU arrival, which potentially would alter the SCAI proportions and outcome, however as we were interested in the hemodynamic course over time and the outcome in relation to vasoactive strategy, installation at CICU was necessary to evaluate this. The treatment target of MAP $> 65$ mmHg and $ScVO_2 > 55\%$ might be to simplified, as there might be patients with higher target values, however these values are the lowest accepted values for AMICS in our institutions, why these limits was selected.

## Conclusion

The current study demonstrates that choice and dose of inopressors were associated with cardiogenic shock severity. Hemodynamic treatment targets were achieved for the majority across all SCAI classes, although the extent of vasoactive support to reach the treatment goals was significantly higher for SCAI class D and E. In SCAI class D and E the mortality was high regardless of vasoactive strategy; only in SCAI C, a significant association was found between epinephrine and 30-mortality in unadjusted analysis that was not significant in in multivariable analysis.

## Supporting information

**S1 Fig.** Hemodynamic in terms of A) mean arterial blood pressure B) Venous oxygen saturation during first 48 hours of CICU admission for each SCAI class.
(DOCX)

**S2 Fig. Mean VIS and mean arterial blood pressure over time according to inopressor choice in each SCAI class.** NE: norepinephrine, DA: dopamine, MIX+AD: Epinephrine and norepinephrine and/or dopamine.
(DOCX)

**S3 Fig. Mean heart rate and mean arterial lactate according to inopressor choice in each SCAI class.**
(DOCX)

**S4 Fig. Mean VIS over 48 hours for patients alive 48 hours after CICU admittance.**
(DOCX)

**S5 Fig. Mulitvariate regression on 30-day mortality.**
(DOCX)

**S1 Table. Patients alive until day 3.**
(DOCX)

**S2 Table. Patient characteristics based on the vasoactive strategy within each SCAI group.**
(DOCX)

**S3 Table. Cause of death in each SCAI class.**
(DOCX)

## Author Contributions

**Conceptualization:** Christian Hassager, Lene Holmvang, Lisette Okkels Jensen, Henrik Schmidt, Hanne Berg Ravn, Jacob Eifer Møller.

**Data curation:** Nanna Louise Junker Udesen, Ole Kristian Lerche Helgestad, Jakob Josiassen, Henrik Frederiksen Højgaard, Louise Linde, Jesper Kjaergaard.

**Formal analysis:** Nanna Louise Junker Udesen.

**Funding acquisition:** Jacob Eifer Møller.

**Investigation:** Nanna Louise Junker Udesen, Ole Kristian Lerche Helgestad, Jakob Josiassen.

**Methodology:** Nanna Louise Junker Udesen, Christian Hassager, Hanne Berg Ravn, Jacob Eifer Møller.

**Supervision:** Christian Hassager, Henrik Frederiksen Højgaard, Louise Linde, Jesper Kjaergaard, Lene Holmvang, Lisette Okkels Jensen, Henrik Schmidt, Hanne Berg Ravn, Jacob Eifer Møller.

**Writing – original draft:** Nanna Louise Junker Udesen.

**Writing – review & editing:** Nanna Louise Junker Udesen, Christian Hassager, Jacob Eifer Møller.

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
