## [Decision Letter · Decision Letter 0]

9 Mar 2022

PONE-D-22-01448Vasoactive pharmacological management according to SCAI class in patients with acute myocardial infarction and cardiogenic shockPLOS ONE

Dear Dr. Udesen,

Thank you for submitting your manuscript to PLOS ONE. After careful consideration, we feel that it has merit but does not fully meet PLOS ONE’s publication criteria as it currently stands. Therefore, we invite you to submit a revised version of the manuscript that addresses the points raised during the review process.

We look forward to receiving your revised manuscript.

Kind regards,

Gaetano Santulli, MD

Academic Editor

PLOS ONE

Journal Requirements:

4. Please upload a new copy of Figures 1, 3 and 4 as the detail is not clear. Please follow the link for more information: https://blogs.plos.org/plos/2019/06/looking-good-tips-for-creating-your-plos-figures-graphics/"" https://blogs.plos.org/plos/2019/06/looking-good-tips-for-creating-your-plos-figures-graphics/

Reviewers' comments:

Reviewer's Responses to Questions

**Comments to the Author**

1. Is the manuscript technically sound, and do the data support the conclusions?

Reviewer #1: Partly

Reviewer #2: Yes

Reviewer #3: Yes

2. Has the statistical analysis been performed appropriately and rigorously? 

Reviewer #1: Yes

Reviewer #2: Yes

Reviewer #3: I Don't Know

3. Have the authors made all data underlying the findings in their manuscript fully available?

Reviewer #1: Yes

Reviewer #2: Yes

Reviewer #3: Yes

4. Is the manuscript presented in an intelligible fashion and written in standard English?

Reviewer #1: Yes

Reviewer #2: Yes

Reviewer #3: Yes

5. Review Comments to the Author

Reviewer #1: This is an interesting study dealing with the vasoactive pharmacological management in patients with AMI and cardiogenic shock classified into different severity subgroups according to the SCAI classes C-E. Overall conclusion which is limited by the retrospective nature of the study is that the use of various vasoactive strategies does not have relevant influence on the short-term mortality, except for the SCAI class C subgroup, where the treatment with epinephrine was associated with higher mortality.

I would like to ask authors to address the following questions:

- Were there any baseline differences within each SCAI class which could favor lower or higher mortality in specific subgroups according to the vasoactive drug regimens? In this context, what about patients who did not obtain revascularization (reasons for that?)? How were those patients distributed within each SCAI class according to the vasoactive subgroup?

- Do the authors know the cause of death in different subgroups of patients, especially in the epinephrine SCAI class C population?

- What about patients, in whom inopressors were given first alone and after some time within the first 72h other inopressors were added. Under which inopressor regimens are those patients classified in this study?

- AMI patients with cardiogenic shock were classified according to the SCAI classes and vasoactive therapy within first 72 hours at CICU and the mortality was shown for different subgroups of patients within the next 30 days. Do the authors have data on the vasoactive therapy beyond the first 72 hours, since significant changes in the composition of the drug therapy could have relevant influence on mortality wrongly blaming the initial choice of the vasoactive substances for the outcome.

Reviewer #2: Vasoactive pharmacological management according to SCAI class in patients with

acute myocardial infarction and cardiogenic shock, despite being retrospective it is a good study. It includes important data, from a representative population through a detailed analysis of patients admitted to intensive care, from 2010 to 2017, in 2 centers that provide tertiary cardiac care for a population of 3.8 million people. Treatment with inopressors was evaluated in detail with regard to its indication in relation to the severity of the disease, its response, and the type of drug used. In my opinion, although only two centers were included, it is a quality work, providing important information regarding the treatment of critically ill patients with cardiogenic shock following

acute myocardial infarction.

Reviewer #3: I read with interest the manuscript entitled "Vasoactive pharmacological management according to SCAI class in patients with acute myocardial infarction and cardiogenic shock" by Udesen, et al. This is a retrospective cohort study of 1250 patients with AMICS examining in-hospital mortality as a function of SCAI Shock stage and vasoactive drug use.

I have the following comments and suggestions:

Introduction - it should be noted that there are more data than just the SOAP-II trial that suggest harm with dopamine/epinephrine versus norepinephrine in CS, including other RCT's and meta-analyses of observational studies. The authors should cite the new SCAI Shock Classification (Naidu, JACC 2022).

Methods - a table describing their SCAI Shock Classification would be helpful, as this is a new approach not used in prior studies. Clarity regarding how hypoperfusion was defined for SCAI stage C should be provided, although I assume it is the same as for the definition of CS itself. The authors should cite references regarding their SCAI Shock Classification, if only the consensus statement. Was cardiac arrest part of the SCAI Classification? Were other MCS devices besides ECMO considered in the SCAI Shock Classification? The authors should double check the VIS formula because a) phenylephrine was not included (although if this was not used it is irrelevant) and b) as I recall, the conversion factor for vasopressin is 10000 not 1000. The authors should specify when the VIS was calculated specifically. My personal opinion is that a simple across-groups comparison for SCAI Stages is not ideal, and instead linear or regression across stages would be more appropriate to determine if there were trends across the stages. This may not be necessary for all of Table 1, but should be done for Table 2 and the physiological variables for Table 1. Did the authors perform logistic regression for their mortality endpoints, either before or after adjustment? This seems important considering the differences between groups...For instance, prior analyses have showed that CICU patients who receive NE do better but only after adjusting for VIS (PMID: 34524266), with an interaction between higher VIS and greater benefit of NE. With 1250 patients, it is not appropriate to just report unadjusted associations, particularly considering the statement in the introduction "however, such studies often assume homogeneity in the AMICS population and do not stratify according to the severity of the disease." At the minimum, they should adjust for SCAI stage, MAP, MCS use and VIS +/- lactate but ideally should include multiple other covariates given the number of outcome events observed. This would help to determine whether the observed associations between vasopressor groups and outcome were due to confounding particularly considering that vasopressor choices changed with SCAI stage (stratifying by SCAI stage is a good start but likely inadequate). Each drug (NE, DA, EPI) can be treated as an independent variable and properly adjusted in this manner. Indeed, propensity adjustment would be ideal although if the authors do a good multivariable analysis I am not sure this extra step is truly necessary.

Results/Figures/Tables - for Table 2, are the values reported the means during the CICU course or during a specific time period? The authors should calculate the ratio of MAP to VIS for inclusion in Table 1 and Table 2, this indexes the BP to the vasopressor load and should be lower in higher SCAI stages. The authors should report the maximum VIS, which has been previously validated as a mortality risk factor in the CICU even when adjusted for other relevant markers (reference #18 plus PMID: 32180344) and in patients with CS (PMID: 33590998 & PMID: 29463462, among others). The authors should also report the maximum # vasoactive drugs, which has been described previously as a marker of prognosis as well. Throughout, the authors should be clear about the time point they are referring to--at CICU admission versus peak, etc. The figures are difficult to read and the size/resolution should be improved. Figure 3 doesn't really show any major differences and is not very interesting. I suggest plotting the MAP/VIS ratio instead if possible and making this supplementary. For Figure 4, both should be line graphs.

Discussion - an important question when considering the ideal vasopressor for CS is whether some drugs are beneficial or other drugs are harmful. My opinion is that NE is safer due to less toxicity than DA/EPI, as supported by studies such as SOAP-II and OptimaCC. Knowing that EPI was used primarily as rescue therapy in this cohort, the authors should discuss whether EPI is directly harmful or whether NE is simply safer. I worry that the results of this and other observational studies not correlating with RCTs, which suggests that there could be confounding by indication--for instance, fewer arrhythmias with DA may imply that it was used selectively in patients with a lower risk of arrhythmias. The DA dose issue may also be true, as the authors astutely note--this is my own experience.

6. PLOS authors have the option to publish the peer review history of their article (what does this mean?). If published, this will include your full peer review and any attached files.

Reviewer #1: No

Reviewer #2: No

Reviewer #3: No

---

## [Author Response · Author response to Decision Letter 0]

31 May 2022

Rebuttal letter

Dear Academic Editor Gaetano Santulli and reviewers. Thank you for giving constructive feedback, we have edited the manuscript based on the comments raised by the editor and the reviewer comments. We believe this has improved the manuscript.

Answer: The new version of the manuscript should comply with the PLOS ONE requirements

Answer: The Funding information has been edited.

Answer: It is not possible to give public access to the data as this would be against the Danish legislation. Permission can only be given individually and is granted by the Danish Data Protection Agency. The same situation was given with the last manuscript my research group published in PLOS ONE (https://doi.org/10.1371/journal.pone.0244294)

4. Please upload a new copy of Figures 1, 3 and 4 as the detail is not clear. Please follow the link for more information: https://blogs.plos.org/plos/2019/06/looking-good-tips-for-creating-your-plos-figures-graphics/"" https://blogs.plos.org/plos/2019/06/looking-good-tips-for-creating-your-plos-figures-graphics/

Answer: The figures have been edited and should be in better format now

Response to reviewers 

Reviewer #1: This is an interesting study dealing with the vasoactive pharmacological management in patients with AMI and cardiogenic shock classified into different severity subgroups according to the SCAI classes C-E. Overall conclusion which is limited by the retrospective nature of the study is that the use of various vasoactive strategies does not have relevant influence on the short-term mortality, except for the SCAI class C subgroup, where the treatment with epinephrine was associated with higher mortality.

I would like to ask authors to address the following questions:

1) Were there any baseline differences within each SCAI class which could favor lower or higher mortality in specific subgroups according to the vasoactive drug regimens? In this context, what about patients who did not obtain revascularization (reasons for that?)? How were those patients distributed within each SCAI class according to the vasoactive subgroup?

Answer: We thank the expert reviewer for this question, and a relevant consideration. In SCAI class C there was greater heterogeneity in relation to the choice of vasoactive strategy, while in SCAI D and E there were less heterogenity. This may explain the higher mortality when using epinephrine in SCAI class C, as this group had a more unfavorable phenotype based on the need for mechanical circulatory support (MCS), mechanical ventilation and hemometabolic status at initial presentation. To describe this, summary information for each SCAI group has been added in appendix, S.6. Mortality rate was higher if revascularization was not performed, which may be due not obvious culprit or severe 3-vessel disease, where CABG first are performed when the cerebral status is known in the OHCA patient or due to relinquishment due to moribund status. There was no significant difference in the proportion who were not revascularized in each SCAI group, appendix S.6. 

2) Do the authors know the cause of death in different subgroups of patients, especially in the epinephrine SCAI class C population?

Answer: We have recorded causes of death for the vast majority of patients, and they are added in Appendix S.8. In general, an increasing proportion dies from cardiac cause with increasing SCAI class, while cerebral anoxia was the most common cause of death in SCAI class C. However, the primary cause of death for those who received epinephrine was cardiac failure regardless of SCAI class.

3) What about patients, in whom inopressors were given first alone and after some time within the first 72h other inopressors were added. Under which inopressor regimens are those patients classified in this study?

Answer: Thank you for the question. We chose the use of inodilators within 24 hours, as we considered this to be a stage where there often is a critical need to increase cardiac output and at the same time hypotension. A stage where safety of inodilators can be raised. In our institutions, levosimendan is frequent at a later stage and also often to wean from MCS, which potentially could influence the outcome, when looking at the later use of inodilators. Inclusion of changes in vasoactive treatment at later stage could also introduce an immortal time bias given the extensive mortality the first 24 hours. 

4) AMI patients with cardiogenic shock were classified according to the SCAI classes and vasoactive therapy within first 72 hours at CICU and the mortality was shown for different subgroups of patients within the next 30 days. Do the authors have data on the vasoactive therapy beyond the first 72 hours, since significant changes in the composition of the drug therapy could have relevant influence on mortality wrongly blaming the initial choice of the vasoactive substances for the outcome.

Answer) A good consideration, and the later use of a certain vasoactive strategy, of course, could also influence the 30-day mortality. We have the data, but chose in this study to focus on the initial treatment at the onset of CS where mortality is highest, and as this is often is where the patients are most hemodynamically compromised. Cause of death also changes dramatically over the initial days (Davodian L et al AJC 2022) where death initial 24-48 hrs is driven by hemodynamic collapse and cardiac failure, whereas later death is mostly related to brain injury after cardiac arrest. To adjust for this a landmark analysis should be performed for patients surviving first 48 hrs, this would add significantly to the complexity of this study and power in this group is low due high initial mortality, and has not been done. Finally, later choice of vasopressors could be influenced by adjacent complications associated with procedures such as cardiac surgery or triggered by sepsis, which could introduce bias. The probability of adverse events increases the longer course we look at, so to decrease the heterogeneity over time we only looked at vasoactive treatment at the initial days. 

Reviewer #2: Vasoactive pharmacological management according to SCAI class in patients with acute myocardial infarction and cardiogenic shock, despite being retrospective it is a good study. It includes important data, from a representative population through a detailed analysis of patients admitted to intensive care, from 2010 to 2017, in 2 centers that provide tertiary cardiac care for a population of 3.8 million people. Treatment with inopressors was evaluated in detail with regard to its indication in relation to the severity of the disease, its response, and the type of drug used. In my opinion, although only two centers were included, it is a quality work, providing important information regarding the treatment of critically ill patients with cardiogenic shock following

acute myocardial infarction.

Answer: 

Thank you, we appreciate your comment.

Reviewer #3: I read with interest the manuscript entitled "Vasoactive pharmacological management according to SCAI class in patients with acute myocardial infarction and cardiogenic shock" by Udesen, et al. This is a retrospective cohort study of 1250 patients with AMICS examining in-hospital mortality as a function of SCAI Shock stage and vasoactive drug use.

I have the following comments and suggestions:

1) Introduction - it should be noted that there are more data than just the SOAP-II trial that suggest harm with dopamine/epinephrine versus norepinephrine in CS, including other RCT's and meta-analyses of observational studies. The authors should cite the new SCAI Shock Classification (Naidu, JACC 2022).

Answer: 

Thank you, we appreciate the in-depth review and excellent comments and suggestion, and we have tried to improve the manuscript based on the suggestions. 

Regarding other studies suggesting harm with dopamine/epinephrine, this has been added in the introduction 

“Guidelines recommend using norepinephrine (NE) as first inopressor for hypotension in AMICS (7,8), primarily based on SOAP-II trial comparing first line NE and dopamine and the CAT study comparing first line NE with epinephrine (9,10). Both studies studied mixed populations of critically ill patients and even though the studies were neutral on primary endpoint (30 day mortality), safety concerns were raised which is in agreement with smaller randomized controlled trials and observational studies suggesting more harm with epinephrine and dopamine” and cited. 

The new SCAI Shock Classification has been cited in the introduction. 

2) Methods - a table describing their SCAI Shock Classification would be helpful, as this is a new approach not used in prior studies. Clarity regarding how hypoperfusion was defined for SCAI stage C should be provided, although I assume it is the same as for the definition of CS itself. The authors should cite references regarding their SCAI Shock Classification, if only the consensus statement. Was cardiac arrest part of the SCAI Classification? Were other MCS devices besides ECMO considered in the SCAI Shock Classification?

Answer: 

Thank you for the comment, in the method section a table (Table 1) describing our SCAI classification has been added. Further inserted in the same section that the SCAI classification are an interpretation based on the original SCAI consensus statement. We chose not to differentiate according to OHCA as we already in the study design according to vasoactive strategy had many subgroups. We have inserted summary information in Appendix S.6, regarding the distribution of OHCA in the different SCAI groups according to the vasoactive strategy. Further, a figure illustrating the effects on 30-days mortality in each SCAI group after multiple regression has been inserted in the appendix, S.7. This shows that after adjusting, OHCA had a significantly higher mortality among SCAI C, but not in the more severe stages of CS. We only considered the early use of VA-ECMO to be extreme stages, as the DanGer Shock trial has been going on in the two institutions in Denmark since early 2013 and thereby may have driven the choice of early Impella use in some patients. 

3) The authors should double check the VIS formula because a) phenylephrine was not included (although if this was not used it is irrelevant) and b) as I recall, the conversion factor for vasopressin is 10000 not 1000. The authors should specify when the VIS was calculated specifically. 

Answer:

A good consideration as phenylephrine could also affect the results. Immediately, none of the AMICS patients received it as a continuous infusion on CICU during the first 3 days. However, we cannot rule out that some patients may have received a bolus phenylephrine during initial management in the catheterization laboratory. We have used the VIS formula from Koponen et all (doi: 10.1016/j.bja.2018.12.019) and here the conversion factor for vasopressin is 10,000. The 1000 it is a typing error and has been corrected - thank for the notification. However, all over the use of vasopressin was scarce. 

4) My personal opinion is that a simple across-groups comparison for SCAI Stages is not ideal and instead linear or regression across stages would be more appropriate to determine if there were trends across the stages. This may not be necessary for all of Table 1, but should be done for Table 2 and the physiological variables for Table 1. Did the authors perform logistic regression for their mortality endpoints, either before or after adjustment? This seems important considering the differences between groups...For instance, prior analyses have showed that CICU patients who receive NE do better but only after adjusting for VIS (PMID: 34524266), with an interaction between higher VIS and greater benefit of NE. With 1250 patients, it is not appropriate to just report unadjusted associations, particularly considering the statement in the introduction "however, such studies often assume homogeneity in the AMICS population and do not stratify according to the severity of the disease." At the minimum, they should adjust for SCAI stage, MAP, MCS use and VIS +/- lactate but ideally should include multiple other covariates given the number of outcome events observed. This would help to determine whether the observed associations between vasopressor groups and outcome were due to confounding particularly considering that vasopressor choices changed with SCAI stage (stratifying by SCAI stage is a good start but likely inadequate). Each drug (NE, DA, EPI) can be treated as an independent variable and properly adjusted in this manner. Indeed, propensity adjustment would be ideal although if the authors do a good multivariable analysis I am not sure this extra step is truly necessary.

Answer: 

Thank you for these insightful comments on the statistical method. We have retained the current analysis in the Tables but were very much in doubt, but we believe the linear regression would assume linearity between the different SCAI classes, this applies in part to SCAI C and E, but as there may be a development to ex SCAI D, why we did not believe that the linear regression was optimal. However it is quite correctly pointed out, that we lack an adjusted analysis for the 30-days mortality analysis. As suggested, we have included a table 5, which shows the odds ratio for death both unadjusted based on SCAI class, but also by multivariate analysis, where the independent variables are the various vasoactive substances, age, maximum VIS, out-of hospital cardiac arrest (OHCA), initial lactate, revascularization, and renal replacement therapy as these variable was associated with outcome, demonstrated in appendix S7, we also chose to include MCS in the multivariate analysis as we thought it could interact. 

5) Results/Figures/Tables - for Table 2, are the values reported the means during the CICU course or during a specific time period? The authors should calculate the ratio of MAP to VIS for inclusion in Table 1 and Table 2, this indexes the BP to the vasopressor load and should be lower in higher SCAI stages. The authors should report the maximum VIS, which has been previously validated as a mortality risk factor in the CICU even when adjusted for other relevant markers (reference #18 plus PMID: 32180344) and in patients with CS (PMID: 33590998 & PMID: 29463462, among others). The authors should also report the maximum # vasoactive drugs, which has been described previously as a marker of prognosis as well. Throughout, the authors should be clear about the time point they are referring to--at CICU admission versus peak, etc. The figures are difficult to read and the size/resolution should be improved. Figure 3 doesn't really show any major differences and is not very interesting. I suggest plotting the MAP/VIS ratio instead if possible and making this supplementary. For Figure 4, both should be line graphs.

Answer:

Thank you for pointing out that it was not clearly described that the values reported in the first part of Table 2 are CICU arrival values. This has now been clarified with an extra heading in the Table 2. The MAP / VIS ratio has been inserted in table 2, however this is not possible in table 1, as we do not have the exact dosages of vasoactive agents before arrival at CICU. We agree that maximum VIS is important in relation to prognosis. The maximum VIS is presented in Table 3, both for all SCAI groups and according to vasoactive strategy. The maximum dose of each vasopressor is presented in Table 2 for each SCAI group. In addition, a multivariate analysis has been inserted in appendix S.7, which include the maximum VIS in each individual SCAI group and the effect on 30-day mortality. As suggested Figure 3 has been replaced with a figure demonstrating the MAP / VIS ratio, as this illustrates the connection between VIS and MAP over time. To show that all groups were able to obtain MAP and SVO2 (however at a different supporting level), Figure 4 is retained in appendix material, S1. Figure 4 has been modified, so that both are line graphs.

6) Discussion - an important question when considering the ideal vasopressor for CS is whether some drugs are beneficial or other drugs are harmful. My opinion is that NE is safer due to less toxicity than DA/EPI, as supported by studies such as SOAP-II and OptimaCC. Knowing that EPI was used primarily as rescue therapy in this cohort, the authors should discuss whether EPI is directly harmful or whether NE is simply safer. I worry that the results of this and other observational studies not correlating with RCTs, which suggests that there could be confounding by indication--for instance, fewer arrhythmias with DA may imply that it was used selectively in patients with a lower risk of arrhythmias. The DA dose issue may also be true, as the authors astutely note--this is my own experience.

Answer 

We agree that the best vasopressor in the treatment of AMICS should be one that provides the best hemodynamic support with the lowest cost in terms of increasing myocardial energy consumption as well as avoiding excessive vasoconstriction and tachycardia leading to ischemia and arrythmia. In terms of vasopressor choice, norepinephrine for the homogeneous group of AMICS is probably preferable, but to differentiate in severity and hemodynamics, low dose dopamine may be suitable for bradycardia in less vasopressor dependent patients and epinephrine for the severe stages. The risk of arrhythmias increases with increasing doses, where with high support, norepinephrine is probably safer as the SOAP-II study demonstrated. In relation to epinephrine, large studies are lacking in AMICS, but with the material currently available, norepinephrine should be the first choice. The appendix material S.6 and the VIS score (Table 3) demonstrate that patients who solely received dopamine in SCAI class C was not needing a high level of support and for those selected patients, dopamine in a low-moderate dose was not harmful. We have added a discussion on the optimal vasopressor in the end of the discussion page 17.

---

## [Decision Letter · Decision Letter 1]

18 Jul 2022

Vasoactive pharmacological management according to SCAI class in patients with acute myocardial infarction and cardiogenic shock

PONE-D-22-01448R1

Dear Dr. Udesen,

We’re pleased to inform you that your manuscript has been judged scientifically suitable for publication and will be formally accepted for publication once it meets all outstanding technical requirements.

Kind regards,

Gaetano Santulli, MD

Academic Editor

PLOS ONE

Reviewers' comments:

Reviewer's Responses to Questions

**Comments to the Author**

1. If the authors have adequately addressed your comments raised in a previous round of review and you feel that this manuscript is now acceptable for publication, you may indicate that here to bypass the “Comments to the Author” section, enter your conflict of interest statement in the “Confidential to Editor” section, and submit your "Accept" recommendation.

Reviewer #1: All comments have been addressed

2. Is the manuscript technically sound, and do the data support the conclusions?

Reviewer #1: Yes

3. Has the statistical analysis been performed appropriately and rigorously? 

Reviewer #1: Yes

4. Have the authors made all data underlying the findings in their manuscript fully available?

Reviewer #1: Yes

5. Is the manuscript presented in an intelligible fashion and written in standard English?

Reviewer #1: Yes

6. Review Comments to the Author

Reviewer #1: Thank you very much for all your answers, including the addition of new data. I agree with your comments and do not have further questions.

7. PLOS authors have the option to publish the peer review history of their article (what does this mean?). If published, this will include your full peer review and any attached files.

Reviewer #1: No

---

## [Editor Report · Acceptance letter]

25 Jul 2022

PONE-D-22-01448R1 

Vasoactive pharmacological management according to SCAI class in patients with acute myocardial infarction and cardiogenic shock 

Dear Dr. Udesen:

I'm pleased to inform you that your manuscript has been deemed suitable for publication in PLOS ONE. Congratulations! Your manuscript is now with our production department. 

Kind regards, 

on behalf of

Professor Gaetano Santulli 

Academic Editor

PLOS ONE